

# Secretomic changes of amyloid beta peptides on Alzheimer's disease related proteins in differentiated human SH-SY5Y neuroblastoma cells

Sittiruk Roytrakul[1], Janthima Jaresitthikunchai[1], Narumon Phaonakrop[1], Sawanya Charoenlappanit[1], Siriwan Thaisakun[1], Nitithorn Kumsri[2] and Teerakul Arpornsuwan[3]

[1] Functional Proteomics Technology Laboratory, National Center for Genetic Engineering and Biotechnology, National Science and Technology Development Agency, Pathumthani, Thailand
[2] Undergraduate Student of Department of Medical Technology, Faculty of Allied Health Sciences, Thammasat University, Pathumtani, Thailand
[3] Medical Technology Research and Service Unit, Health Care Service Center, Faculty of Allied Health Sciences, Thammasat University, Pathumthani, Thailand

Corresponding author
Teerakul Arpornsuwan,
teerakul@tu.ac.th

## ABSTRACT

Alzheimer's disease (AD) is a neurodegenerative disease that causes physical damage to neuronal connections, leading to brain atrophy. This disruption of synaptic connections results in mild to severe cognitive impairments. Unfortunately, no effective treatment is currently known to prevent or reverse the symptoms of AD. The aim of this study was to investigate the effects of three synthetic peptides, *i.e.*, KLVFF, RGKLVFFGR and RIIGL, on an AD *in vitro* model represented by differentiated SH-SY5Y neuroblastoma cells exposed to retinoic acid (RA) and brain-derived neurotrophic factor (BDNF). The results demonstrated that RIIGL peptide had the least significant cytotoxic activity to normal SH-SY5Y while exerting high cytotoxicity against the differentiated cells. The mechanism of RIIGL peptide in the differentiated SH-SY5Y was investigated based on changes in secretory proteins compared to another two peptides. A total of 380 proteins were identified, and five of them were significantly detected after treatment with RIIGL peptide. These secretory proteins were found to be related to microtubule-associated protein tau (MAPT) and amyloid-beta precursor protein (APP). RIIGL peptide acts on differentiated SH-SY5Y by regulating amyloid-beta formation, neuron apoptotic process, ceramide catabolic process, and oxidative phosphorylation and thus has the potentials to treat AD.

## INTRODUCTION

Alzheimer's disease (AD) is a dynamic neurological disorder that accounts for the most noteworthy number of dementia cases. The pathophysiology of AD is basically caused by aggregation and accumulation of amyloid-β (Aβ) plaques and hyperphosphorylated tau proteins within the brain. The accumulation of the aggregated proteins is considered

the pathological hallmarks of AD (*Mhillaj et al,, 2020*). Five drugs, Tacrine, Donepezil, Carbalatine, Galanthamine, and Memantine, have been approved by the Food and Drug Administration (FDA) for clinical use. Anti-Aβ antibodies, such as donanemab, lecanemab, and aducanumab, have shown promise in reducing Aβ levels and decelerating cognitive decline (*Haddad et al., 2022*; *Jucker & Walker, 2023*; *Sims et al., 2023*; *van Dyck et al., 2023*). Lecanemab received approval in 2023 for AD treatment. However, these drugs only manage symptoms and delay the onset of AD but do not cure it. In addition, several drugs are undergoing clinical trials for AD, but unfortunately, many have been terminated because of poor efficacy or large adverse reactions.

Peptide-based therapeutics have garnered significant attention in recent years due to their high affinity for Aβ and low toxicity (*Danho et al., 2009*). Among Aβ aggregates, the Aβ oligomers are particular toxic, thus, targeting oligomerization process of Aβ offers an optimal approach with minimal associated risks. Certain peptides derived from amyloid beta effective inhibit Aβ aggregation. These include KLVFF, RGKLVFFGR and RIIGL peptides.

KLVFF peptide containing a self-recognition component specific to Aβ42 has been found to minimize the β-strand hydrophobic core region of the Aβ peptide. The KLVFF pentapeptide interacts with monomers and oligomers of Aβ42 with high specificity, leading to inhibition of fibril formation, with no cytotoxic effect (*Horsley et al., 2020*).

RGKLVFFGR peptide contains the central region of Aβ (residues 16–20) and plays important role in inhibition of Aβ aggregate formation. It has high efficacy to block the neurotoxicity of β-amyloid in human neuroblastoma cells. The formations of both β-amyloid oligomers and fibrils were prevented by this peptide (*Matharu et al., 2010*).

RIIGL pentapeptide is a derivative of the beta-amyloid 31-34 (Abeta 31–34) and exhibits no toxicity against neuroblastoma culture. After co-incubation with Abeta (1–42), it inhibits the formation of amyloid fibers and lowers the cytotoxic effect of fibrillar Abeta (1–42). These results demonstrate that RIIGL peptide is the novel neuroprotective peptidomimetics which can be an effective inhibitor of both the aggregation and the toxic effects of amyloid-β (1–42) (*Fülöp et al., 2004*).

Neuronal cells, SH-SY5Y, upon sequential exposure of retinoic acid (RA) and brain-derived neurotrophic factor (BDNF), differentiated into a cholinergic phenotype which is suitable for AD based studies (*Agholme et al., 2010*; *de Medeiros et al., 2019*). Recently, the impact of amyloid beta peptide derivatives on anti-amyloid cytotoxicity activity and the proteins involved in growth inhibition of the differentiated SH-SY5Y cells has not yet been investigated. The aim of this study is to investigate the effect of the three peptides, *i.e.*, KLVFF, RGKLVFFGR and RIIGL, on expressions of proteins related with β-amyloid and tau protein that causing AD in the differentiated SH-SY5Y cells induced by RA and BDNF.

## MATERIALS & METHODS

### Peptides

KLVFF, RGKLVFFGR and RIIGL peptides were chemically synthesized by a solid-phase method and purified by HPLC with higher purities than 95% (GenScript, Piscataway,

NJ, USA). All synthetic peptides were dissolved in $\geq$ 99.5% dimethyl sulfoxide (DMSO; Sigma, Saint-Quentin-Fallavier, France) and further diluted in fresh antibiotic-free culture medium to obtain the required working concentrations.

## Cell lines and cell culture

Vero cell line (ATCC CCL-81) was used for *in vitro* cytotoxicity assay. Cell viability assay was studied using human neuroblastoma cells, SH-SY5Y (ATCC CRL-2266, lot Number 70047995). Both cell lines were commercially obtained from ATCC (Manassas, VA, USA). Cells were grown in Dulbecco's Modified Eagle Medium (DMEM; Sigma) supplemented with 10% inactivated fetal bovine serum (FBS) (Gibco), 100 units per milliliter of penicillin, and 100 $\mu$g/mL of streptomycin (Invitrogen, USA). All cultures were kept alive in a humidified 37 °C incubator with 5% $CO_2$ - 95% air environment. Every 3 to 4 days, the medium was replaced, and the cells were subcultured as necessary.

## *In vitro* cytotoxicity assay of Vero cells

KLVFF, RGKLVFFGR, and RIIGL were tested for *in vitro* cytotoxicity against Vero cells using the colorimetric MTT assay, as previously described by *Arpornsuwan et al. (2014)*. The experiments were done in triplicates. Vero cells in the logarithmic growth phase were collected by trypsinization, washed with PBS, and resuspended in fresh media containing 10% FBS. The cells at 80–90% confluence containing $1 \times 10^4$ cells were plated in each well of 96 well tissue culture plate. The cells were then incubated at 37 °C for an overnight period with 5% $CO_2$ and 95% air. The cells were treated with KLVFF, RGKLVFFGR or RIIGL at concentration of 100 $\mu$g/mL followed by cell proliferation assay. The MTT solution at concentration of 1 mg/mL was replaced in each well. The formazan crystals were dissolved in 100 $\mu$L of DMSO after incubation under 5% $CO_2$ and 95% air at 37 °C for 3 h. The absorbance was recorded at 590 nm using a microtiter plate reader.

## Hemolytic activity against human red blood cells

The hemolytic activity of KLVFF, RGKLVFFGR, RIIGL against human red blood cells was measured as previously described by *Arpornsuwan et al. (2014)*. Briefly, one milliliter of red cells was added into 100 mL of PBS (1% cell suspension). Each well of the microtiter plate with different peptides at concentration of 100 $\mu$g/mL was mixed with 50 $\mu$L of 1% red cell suspension. The microtiter plate was subsequently incubated for 3 h at 37 °C. After incubation, the microtiter plate was centrifuged at 1,700$\times$ g for 5 min, and the optical density of the supernatant was determined at 590 nm. As a positive control, 1% SDS in distilled water was utilized, and the PBS solution served as a negative control. The hemolysis percentage was calculated as previously described by *Arpornsuwan et al. (2014)*. The experiments were performed in three replicates.

## Differentiation of SH-SY5Y cells and cytotoxicity test

The SH-SY5Y cells were differentiated into mature neurons with morphological characteristic of mature cells using retinoic acid (RA) and BDNF as a preferable model for neuroscience with a modified method as described by *Jämsä et al. (2004)* and *Riegerová et al. (2021)*. SH-SY5Y cells were seeded at $5 \times 10^3$ cells/well in 96-well plates containing
DMEM supplemented with 10% heat-inactivated fetal bovine serum and incubated under 5% $CO_2$ and 95% air at 37 °C for 72 h. Then 10 µM of retinoic acid (RA) was added to cultured cells and incubated for overnight. The cells were washed with DMEM medium and 50 ng/mL of Brain-derived neurotrophic factor (BDNF) was added into the cells and incubated for 48 h, followed by washing step with the medium. Finally, the differentiated cells were treated with 100 µg/mL of each peptide, then incubated for overnight. The undifferentiated SH-SY5Y cells cultured in medium for 72 h without treatment with RA and BDNF were used as cell control. After peptide treatment, cell viability was measured by MTT assay according to *Arpornsuwan et al. (2014)*. The experiments were conducted in three replicates.

## Proteomic analysis

The cell culture medium samples were mixed with an equal volume of cold acetone and incubated at −20 °C overnight. The samples were then centrifuged at 10,000 rpm for 30 min. The supernatant was discarded and the protein pellet was dissolved in 0.1% SDS. The protein concentration of the total proteins isolated from the SH-SY5Y culture medium was determined using the Lowry assay with bovine serum albumin (BSA) as a standard protein (*Lowry et al., 1951*). Five micrograms of protein samples were reduced with 5 mM dithiothreitol at 60 °C for 1 h. The samples were then alkylated with 15 mM iodoacetamide at room temperature for 45 min in the dark. The protein samples were digested with sequencing grade porcine trypsin (at a 1:20 ratio) for 16 h at 37 °C. The tryptic peptide solutions were dried using a speed vacuum concentrator before storage at −20 °C until analysis.

The tryptic peptide samples were resuspended in 0.1% formic acid and analyzed by liquid chromatography-mass spectrometry (LC-MS) on an Ultimate3000 Nano/Capillary LC System (Thermo Scientific, Waltham, MA, USA) coupled to a ZenoTOF 7600 mass spectrometer (SCIEX, Framingham, MA, USA). DDA method selection of the top, most abundant top 50 precursor ions per survey MS1 for MS/MS at an intensity threshold exceeding 150 cps. Sampled precursor ions were dynamically excluded for 12 s after two incidences of MS/MS sampling occurrence (MS/MS sampling with dynamic CE for MS/MS enabled). The MS2 spectra were collected from 100–1,800 m/z with a 50-ms accumulation time and Zeno trap enabled. To minimize the effect of experimental variation, three independent MS/MS runs were performed for each sample.

MaxQuant 2.1.4.0 was used to quantify and identify the proteins in individual samples using the Andromeda search engine to correlate raw MS/MS spectra to the known proteins related to AD downloaded from Uniprot *Homo sapiens* database (*Tyanova, Temu & Cox, 2016*). The MaxQuant's standard setting parameters used were enzyme (trypsin), fixed modification (carbamidomethylation of cysteine residues), variable modifications (oxidation of methionine residues and acetylation of the protein N-terminus) and two missed cleavages. Protein FDR was set at 1% and estimated by the reversed search sequences. Gene ontology (GO) enrichment analysis was conducted on differentially expressed proteins using the ShinyGO (version 0.77) (*Ge, Jung & Yao, 2020*). The Venn diagram was constructed by jvenn and showed the identified proteins among the sample groups

(*Bardou et al., 2014*). The visualization and statistical analyses of the differentially expressed proteins, including principal component analysis (PCA), differential analysis (ANOVA and heatmap) were conducted using Metaboanalyst with a significance threshold of *P*-value < 0.05 (*Pang et al., 2022*).

## Statistical analyses

The data were expressed as mean ± standard deviation (SD). Student's *t*-test were performed to conduct statistical analysis (GraphPad Prism 7.0). Microsoft Office Excel 365 was used for data analyses and graphs. Values of $p < 0.05$ were indicative of significant differences.

# RESULTS

## *In vitro* cytotoxicity assay

The cytotoxicity of three synthetic peptides, KLVFF, RGKLVFFGR and RIIGL against Vero cells was investigated as shown in Fig. 1. All studied peptides had low cytotoxic activity on Vero cells by 0.54–2.46% at 100 μg/mL (Fig. 1).

## Hemolytic activity against human red blood cells

The hemolytic activity using human red blood cells was also determined to investigate the selectivity of peptide against the host cells. The results showed that RIIGL had least hemolytic activity, less than 3% at 100 μg/mL of peptide concentration (Fig. 1). While KLVFF and RGKLVFFGR had hemolytic activity by 20–27.26% at 100 μg/mL (Fig. 1).

## Anti-proliferative effect of peptides to SH-SY5Y cell line

The cell proliferations of SH-SY5Y and differentiated SH-SY5Y treated with three synthetic peptides at 100 μg/mL were shown in Fig. 2. All studied peptides inhibited growth of undifferentiated SH-SY5Y with less than 5% cell death. Moreover, RGKLVFFGR and RIIGL at the concentration of 100 μg/mL affected the cytotoxicity 32.43% and 33.21%, respectively, of differentiated SH-SY5Y. In contrast, KLVFF had the least cytotoxic activity against the differentiated SH-SY5Y at 4.21% cell death.

## Proteins in culture media of differentiated SH-SY5Y cells after exposure to three synthetic peptides

Total proteins were extracted from the culture media of SH-SY5Y and differentiated SH-SY5Y cells after exposure to KLVFF, RGKLVFFGR and RIIGL peptides and analyzed using LC-MS. All raw MS/MS data were used to quantify and identify the proteins related to human AD in individual samples using MaxQuant 2.1.4.0. Finally, all data were exported to MetaboAnalyst 5.0 for bioinformatically analysis. All 380 differential proteins were significantly identified ($p < 0.05$) as shown in Fig. 3. Bioinformatic enrichment analyses based on the GO classification system were performed. In the category of biological process, ShinyGO analyses showed that the secretory proteins were mainly represented by proteins of amyloid beta metabolic process, regulation of neuron death by apoptosis or autophagy, neuron development, cell–cell signaling and regulation of transport (Fig. 4).

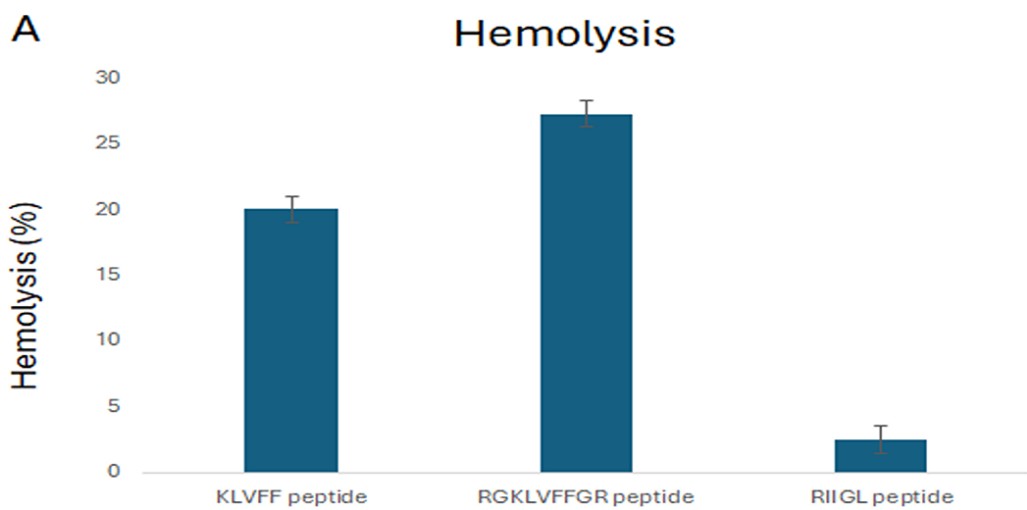

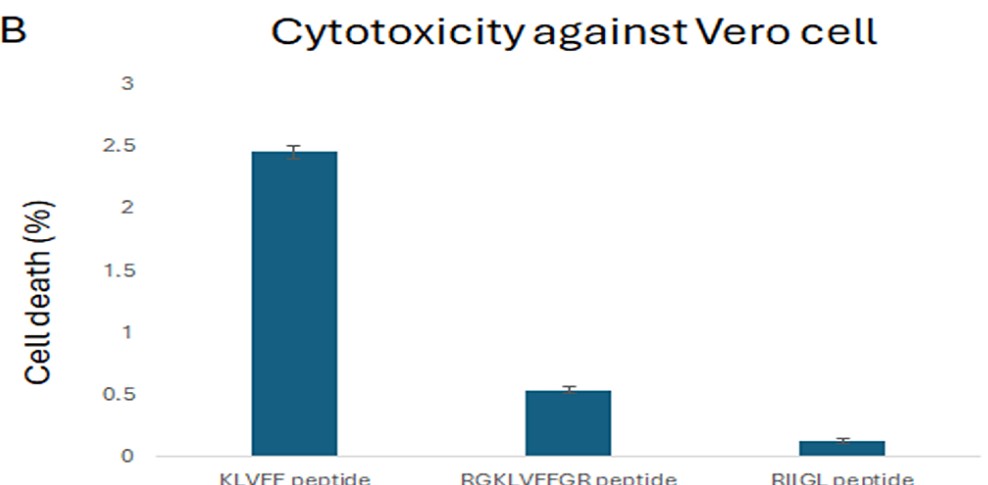

**Figure 1** **(A-B) Percentage of cytotoxicity in Vero cell line and hemolytic activity against human red blood cell.** (A) The hemolytic activity using human red blood cells was determined to investigate the selectivity of peptide against the host cells. (B) The cytotoxicity of three synthetic peptides, KLVF, RGKLVF-FGR and RIIGL against Vero cells was investigated.

Principal component analysis (PCA) score plot showed that there was clear separation of SH-SY5Y and differentiated SH-SY5Y secretomes (Fig. 5). Hierarchical clustering dendrograms also showed that the differentiated SH-SY5Y culture medium was clearly separated from those of the control SH-SY5Y as shown in Fig. 6. To investigate the characteristics of the differentiated SH-SY5Y cells in response to three synthetic peptides, the low dimensional variations (PC1 17.7% and PC2 8.9%) suggested that most proteins related to AD were similar among the treatments. A total of 290, 240, 284, 303, and 290 proteins were identified in the secretomes of SH-SY5Y, differentiated SH-SY5Y, KLVFF treated differentiated SH-SY5Y, RGKLVFFGR treated differentiated SH-SY5Y and RIIGL treated differentiated SH-SY5Y, respectively (Fig. 7). Eight proteins were uniquely detected

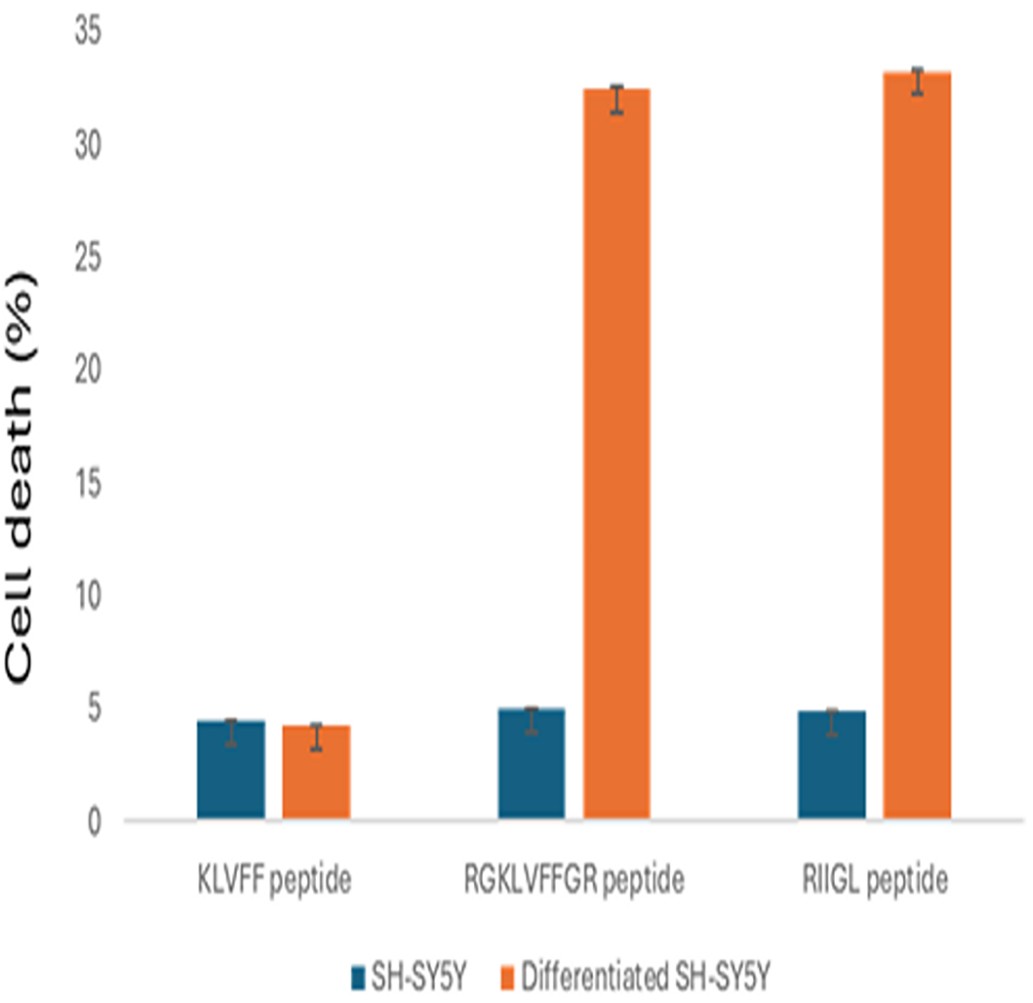

**Figure 2  Anti-proliferative effect of 100 μg/mL peptides on normal SH-SY5Y and differentiated SH-SY5Y cell.** The cell proliferations of SH-SY5Y and differentiated SH-SY5Y treated with three synthetic peptides at 100 μg/mL were investigated.

in normal SH-SY5Y, three proteins in differentiated SH-SY5Y treated with KLVFF peptide, 10 proteins in differentiated SH-SY5Y treated with RGKLVFFGR peptide and 5 proteins in RIIGL treated differentiated SH-SY5Y secretomes (Table 1).

The RIIGL peptide exerted high potential to be used as therapeutics for AD due to less toxicity against normal red blood cells and Vero cells as well as high cytotoxicity against differentiated SH-SY5Y cells. Five proteins uniquely detected in RIIGL treated differentiated SH-SY5Y secretome were selected for further identification of potential proteins related to AD. AD related signaling pathway was predicted using STITCH version 5.0. The interactions between microtubule-associated protein tau (MAPT) and amyloid-beta precursor protein (APP) causing AD and these identified proteins were shown in Fig. 8.

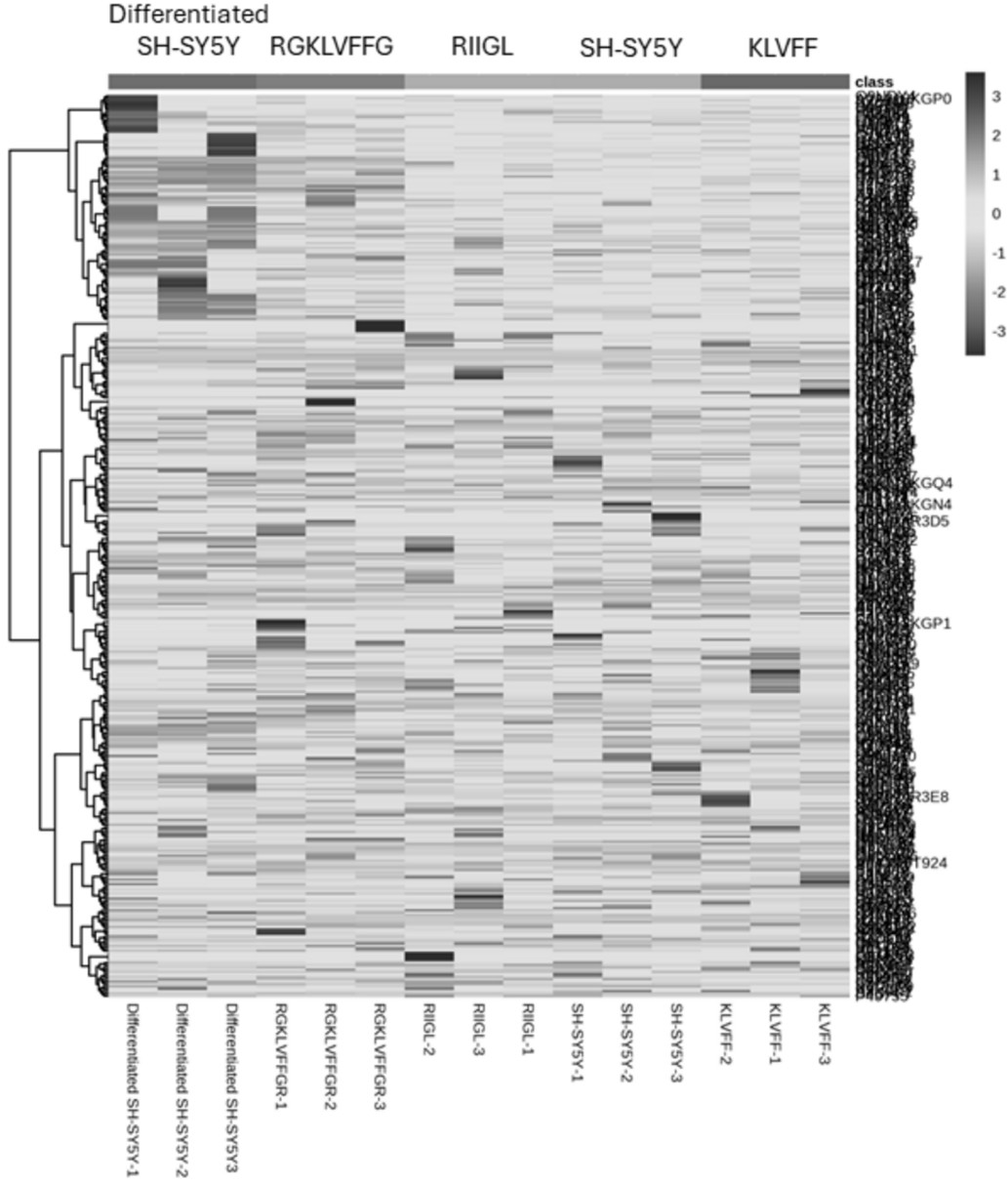

**Figure 3** **A heatmap of significantly and differentially expressed proteins in the secretomes of SH-SY5Y, differentiated SH-SY5Y before and after treatment with three peptides.** LC-MS analysis of each sample was performed in triplicate, resulting in the identification of 380 proteins with varying expression levels across groups. The columns represent samples, while the rows correspond to identified proteins. The color scale indicates proteins intensity, ranging from very low (light grey) to extremely high (black) as depicted on the right side of the heatmap. Abbreviations: SH-SY5Y: undifferentiated SH-SY5Y; differentiated SH-SY5Y: differentiated SH-SY5Y; RGKLVFFGR peptide: differentiated SH-SY5Y treated with RGKLVFFGR peptide; RIIGL peptide: differentiated SH-SY5Y treated with RIIGL peptide; KLVFF peptide: differentiated SY5Y treated with KLVFF peptide.

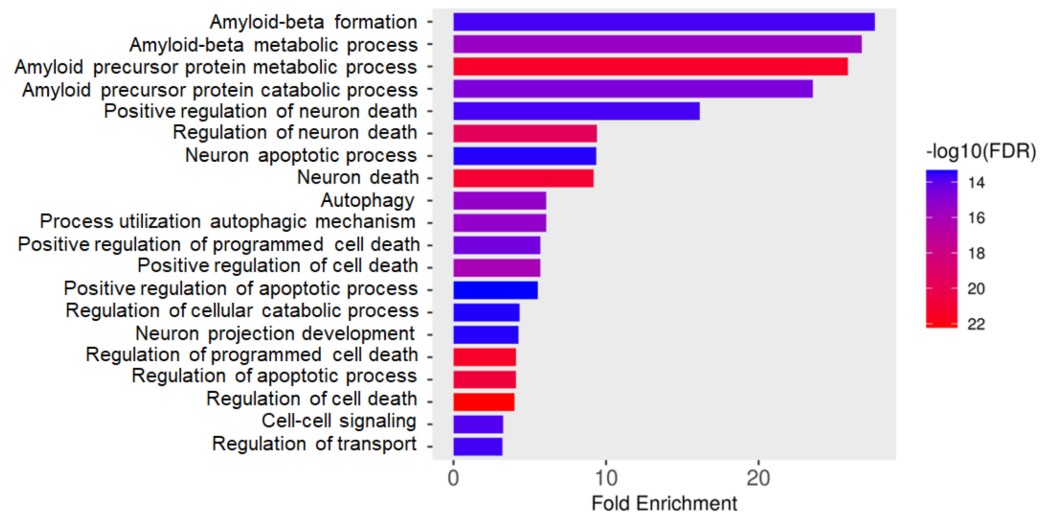

**Figure 4  Gene ontology (GO) distributions of secreted proteins.** Function classification of 380 proteins identified in secretomes of SH-SY5Y, differentiated SH-SY5Y before and after treatment with three peptides.

## DISCUSSION

This study used proteomic analysis to investigate the mechanisms of the action of 3 amyloid beta peptides *in vitro*. SH-SY5Y cells were differentiated into a cholinergic phenotype which is suitable for AD-based studies. Cytotoxicity and hemolytic activity of the peptides were determined. In addition, how the secretion of proteins related to β-amyloid and tau protein in SH-SY5Y cells induced by retinoic acid (RA) and brain-derived neurotrophic factor (BDNF) was affected by peptides was investigated.

Vero cells have been widely used as a control in various research contexts due to their well-documented safety profile and widely acceptance for *in vitro* studies (*Andreani et al., 2017*; *Sheth, 2022*; *Yu et al., 2022*). Using Vero cells as reference point allows researchers to establish a standardized baseline for assessing cytotoxicity. This helps identification whether test substances have a more significant impact on cell viability than the established baseline. In addition, the consistent responses of Vero cells to cytotoxic agents enhance the accuracy of cytotoxicity result interpretation. Researchers assess drug efficacy, safety, and mechanisms using Vero cell-based assays.

Researchers frequently evaluate the efficacy, safety, and mechanisms of anti-Alzheimer's compounds using MTT cytotoxicity assay (*Sehra et al., 2024*; *Sharon et al., 2024*; *Tahir et al., 2024*). This assay serves as a convenient reference point enabling researchers to establish a standardized baseline for assessing cytotoxicity. By comparing the impact of anti-Alzheimer's compounds on cell viability to this established baseline, one can identify their relative effects.

KLVFF and RGKLVFFGR peptides showed high cytotoxicity effect on red blood cells but low cytotoxicity against Vero cells (Fig. 1). RIIGL showed low toxicity to both red blood cells and Vero cells. Low toxicity of KLVFF peptide against neuroblastoma SH-SY5Y

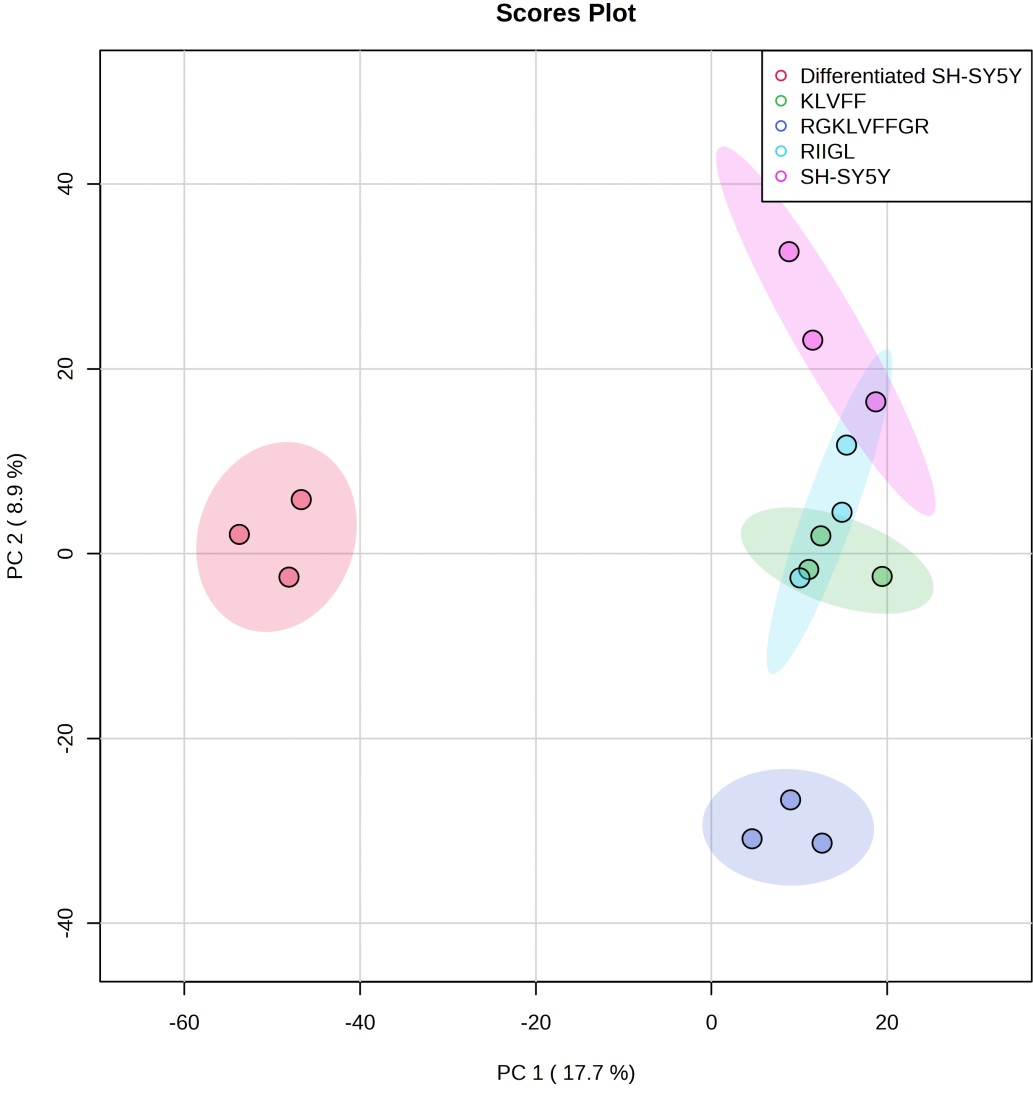

**Figure 5** **Principal component analysis of the secreted proteins.** PCA of the AD related proteins in a 2-dimensional graphs of PC1 and PC2. The biplot shows proteomic data (scores) as labeled dots and treatment with peptides as vectors for the differentiated SH-SY5Y before and after treatment with 3 peptides. Abbreviations: SH-SY5Y: undifferentiated SH-SY5Y; differentiated SH-SY5Y: differentiated SH-SY5Y; RGKLVFFGR peptide: differentiated SH-SY5Y treated with RGKLVFFGR peptide; RIIGL peptide: differentiated SH-SY5Y treated with RIIGL peptide; KLVFF peptide: differentiated SH-SY5Y treated with KLVFF peptide.

cells and differentiated SH-SY5Y AD-based cells were observed. Interestingly, RIIGL and RGKLVFFGR peptides exhibited high toxicity against both normal SH-SY5Y cells and differentiated SH-SY5Y cells. RIIGL had less toxicity against red blood cells and Vero cells but exhibited higher cytotoxicity against differentiated SH-SY5Y than those of KLVFF and RGKLVFFGR. This suggests that RIIGL peptide can be therefore developed as therapeutics for AD.

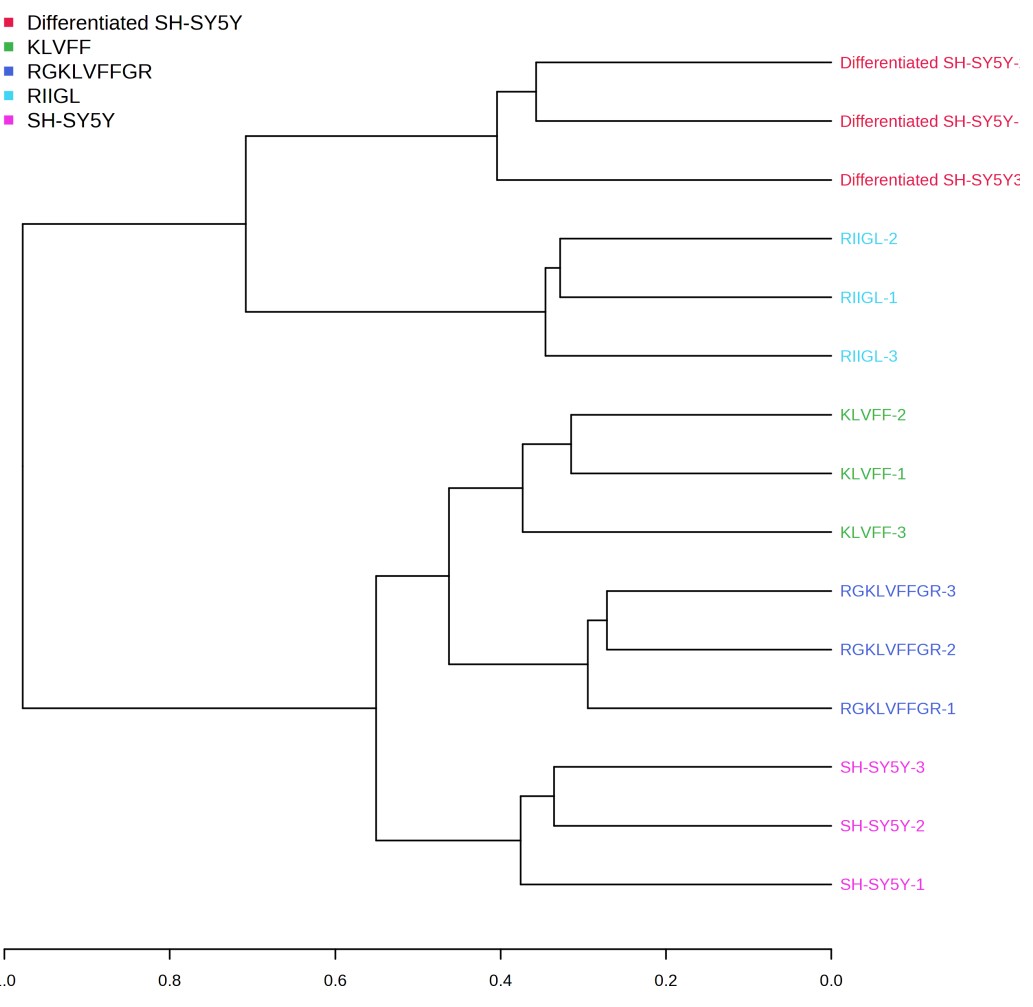

**Figure 6** **Dendrogram of the secreted proteins.** Dendrogram showing the relationship between the AD related proteins in differentiated SH-SY5Y before and after treatment with three peptides using Spearman distances and ward clustering. Abbreviations: SH-SY5Y, undifferentiated SH-SY5Y; differentiated SH-SY5Y, differentiated SH-SY5Y; RGKLVFFGR peptide, differentiated SH-SY5Y treated with RGKLVFFGR peptide; RIIGL peptide, differentiated SY5Y treated with RIIGL peptide; KLVFF peptide, differentiated SY5Y treated with KLVFF peptide.

KLVFF, a small peptide corresponding to the amino acid sequence 16–20 of Aβ can interfere with amyloid aggregation (*Chafekar et al., 2007*; *Phongpradist et al., 2022*; *Trebesova et al., 2022*). The ability of KLVFF peptide to restore the viability of differentiated SH-SY5Y (Fig. 2) is consistent with the anti-aggregation properties of KLVFF derivatives already reported (*Castelletto et al., 2017*; *Tjernberg et al., 1996*; *Tjernberg et al., 1997*; *Wood et al., 1995*).

RGKLVFFGR is a retro-inverse peptide designed by adding arginine and glycine to the KLVFF sequence (*Austen et al., 2008*). It has a high resistance to proteolysis while maintaining the same inhibitory effects on Aβ oligomer formation and cytotoxicity. Interestingly, this peptide has less effect on the viability of normal SH-SY5Y cells but

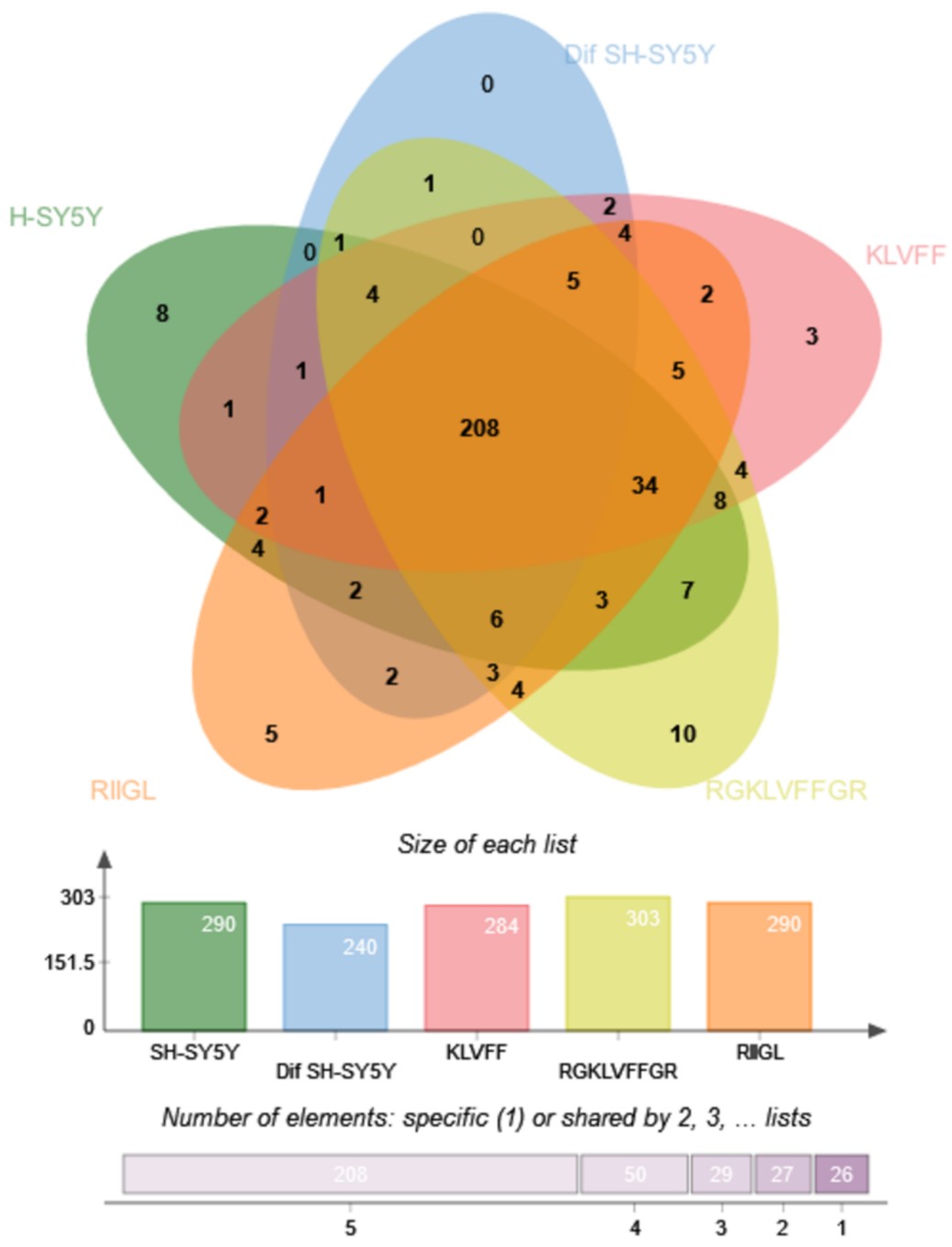

**Figure 7** **Venn diagram of the secreted proteins.** The number of common and exclusive differentially expressed proteins detectable in secretomes of SH-SY5Y, differentiated SH-SY5Y before and after treatment with three peptides were shown. Abbreviations: SH-SY5Y, undifferentiated SH-SY5Y; differentiated SH-SY5Y, differentiated SH-SY5Y; RGKLVFFGR peptide, differentiated SH-SY5Y treated with RGKLVFFGR peptide; RIIGL peptide, differentiated SH-SY5Y treated with RIIGL peptide; KLVFF peptide, differentiated SH-SY5Y treated with the KLVFF peptide.

**Table 1  A list of 26 proteins uniquely detected in normal SH-SY5Y cell and differentiated SH-SY5Y cell treated with KLVFF peptide, RGKLVF-FGR, and RIIGL peptides, respectively.**

| Protein | Protein name | Function | Subcellular distribution |
|---|---|---|---|
| | | Normal SH-SY5Y | |
| Q71HJ6 | Synaptopodin | modulating actin-based shape and motility of dendritic spines and renal podocyte foot processes | cytoplasm |
| B4DFF6 | cDNA FLJ57264 | heme transmembrane transporter activity | endosome, plasma membrane |
| Q54A46 | Amyloid beta (A4) protein-binding, family B, member 3, isoform CRA_g | regulation of DNA-templated transcription | cytoplasm, nucleus |
| Q6FGG5 | Scrapie responsive protein 1, isoform CRA_a | mesenchymal stem cell proliferation | neuron projection terminus |
| B4DFG1 | N(alpha)-acetyltransferase NatA | peptide-glutamate-alpha-N-acetyltransferase activity | NatA complex |
| B4DFJ7 | Tetraspanin 31, isoform CRA_a | positive regulation of cell population proliferation | membrane |
| Q9B2U4 | NADH-ubiquinone oxidoreductase chain 6 | mitochondrial electron transport | mitochondrial inner membrane |
| B5TZ82 | Estrogen receptor (Nuclear receptor subfamily 3 group A member 1) | intracellular estrogen receptor signaling pathway | nucleoplasm |
| | | Differentiated SH-SY5Y cell treated with KLVFF peptide | |
| Q8IWL8 | Saitohin | positive regulation of mRNA splicing | cytoplasm, nucleus |
| Q4VFD5 | Cytochrome b | mitochondrial electron transport | mitochondrial respiratory chain complex III |
| B5TZ79 | Estrogen receptor (Nuclear receptor subfamily 3 group A member 1) | intracellular estrogen receptor signaling pathway | nucleoplasm |
| | | Differentiated SH-SY5Y cell treated with RGKLVFFGR peptide | |
| C9E0F1 | Estrogen receptor (Nuclear receptor subfamily 3 group A member 1) | intracellular estrogen receptor signaling pathway | nucleoplasm |
| B4DFF2 | Sialin | monoatomic anion transport | membrane |
| C8CJL5 | Estrogen receptor (Nuclear receptor subfamily 3 group A member 1) | intracellular estrogen receptor signaling pathway | nucleoplasm |
| Q9Y5Z0 | Beta-secretase 2 | amyloid-beta metabolic process | endoplasmic reticulum, Golgi apparatus, plasma membrane |
| Q5RLM1 | Neuronal acetylcholine receptor alpha-4 subunit | acetylcholine receptor signaling pathway | plasma membrane |
| Q9BXQ0 | Tissue transglutaminase | cAMP-mediated signaling | cytoplasm |
| Q96I26 | PTPN11 protein | Positively regulates MAPK signal transduction pathway | cytoplasm, nucleus |
| C0LJA7 | Estrogen receptor (Nuclear receptor subfamily 3 group A member 1) | intracellular estrogen receptor signaling pathway | nucleoplasm |
| B4DFJ2 | Protein FAM35A | regulation of double-strand break repair | nucleus |
| B7Z1X7 | cDNA FLJ59017 | unknown | unknown |

**Table 1** (*continued*)

| Protein | Protein name | Function | Subcellular distribution |
|---|---|---|---|
| | | Differentiated SH-SY5Y cell treated with RIIGL peptide | |
| GDNF | Glial cell line-derived neurotrophic factor | regulation of neuron apoptotic process | extracellular region |
| TMED10 | Transmembrane emp24 domain-containing protein 10 | regulation of amyloid-beta formation | endoplasmic reticulum, Golgi membrane |
| ASAH2B | Putative inactive neutral ceramidase B | ceramide catabolic process | Golgi apparatus |
| MT-ND4 | NADH-ubiquinone oxidoreductase chain 4 | oxidative phosphorylation | mitochondrial inner membrane |
| NRGN | Neurogranin | nervous system development | axon |

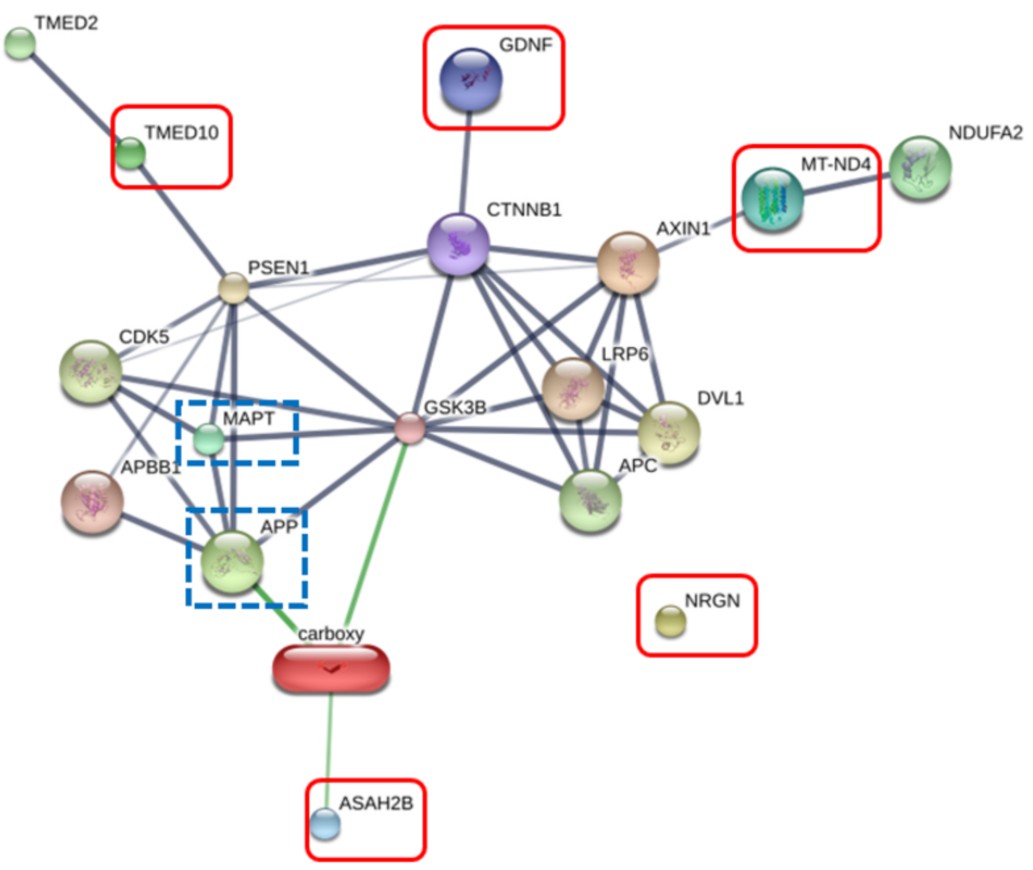

**Figure 8** **Network of the secreted proteins after differentiated SH-SY5Y exposure to RIIGL peptide.** Associations of five proteins identified in the secreted proteins from differentiated SH-SY5Y after treated with RIIGL peptide and AD related proteins including MAPT and APP. Stronger associations are represented by thicker lines. Weak associations are represented by thin lines.

showed a cytotoxic effect against differentiated SH-SY5Y cells, as shown in Fig. 2. In contrast, RGKLVFFGR exhibited less effectiveness on Aβ42 aggregation as reported by

*Austen et al. (2008)*. According to our results, this peptide is a better inhibitor than the KLVFF peptide.

RIIGL peptide acts as a cytotoxic agent in differentiated SH-SY5Y cells but does not display inherent toxicity in normal SH-SY5Y cells, as shown in Fig. 2. This peptide was reported to inhibit the formation of amyloid fibers and decrease the cytotoxic effect of fibrillar Aβ1–42 (*Fülöp et al., 2004*). Taken together, these findings indicate that RIIGL peptide is an effective inhibitor for aggregation and toxic effects of A β1–42 and can serve as a lead candidate for the novel neuroprotective peptide.

Figures 1 and 2 demonstrate that RIIGL, KLVFF, and RGKLVFFGR peptides exhibit high cytotoxicity against differentiated SH-SY5Y cells, which correlates with their ability to inhibit Aβ42 fibril formation and toxicity in an *in vitro* AD model (*Chafekar et al., 2007*; *Phongpradist et al., 2022*; *Trebesova et al., 2022*; *Castelletto et al., 2017*; *Tjernberg et al., 1996*; *Tjernberg et al., 1997*; *Wood et al., 1995*). A neuroprotective effect on SH-SY5Y cytotoxicity induced by retinoic acid (RA) and brain-derived neurotrophic factor (BDNF) could be thus used as an AD *in vitro* model.

Peptides are promising candidates for the treatment of neurodegenerative diseases, owing to their high specificity, low toxicity, and low immunogenicity, which have attracted considerable interest in drug development. However, the applicability of the tested peptides may be limited by their high concentrations. Therefore, RIIGL peptide should be further modified and optimized to enhance its inhibitory effect on A β aggregation and AD. The activity, specificity, stability, toxicity, and immunogenicity of modified RIIGL peptides should be determined *in vitro*. Further evaluation is warranted for the optimal peptide(s) with the lowest effective concentration. This evaluation should encompass efficacy, cytotoxicity, and stability in an animal model. Several *in vivo* studies (*van Groen et al., 2008*; *Maebuchi et al., 2013*; *Katayama et al., 2014*) have provided evidence suggesting that the specific peptides could have beneficial effects in preventing dementia, including AD. Among these peptides, the D-RPRTRLHTHRNR (D3) peptide has demomnstrated high permeability across the blood–brain barrier and high bioavailability after oral administration (*Jiang et al., 2016*).

A shotgun proteomic analysis of secretory proteins in culture media of normal SH-SY5Y and differentiated SH-SY5Y cells after treatment with three synthetic peptides identified 380 proteins. The principal component analysis and dendrogram (Figs. 5 and 6) clearly separated not only the secretomes of normal SH-SY5Y and differentiated SH-SY5Y cells but also the secretomes of differentiated SH-SY5Y cells after exposure to different peptides KLVFF, RGKLVFFGR, and RIIGL. Biological process analyses of all identified proteins revealed enrichment for processes largely involved in amyloid beta metabolic process and regulation of neuron death by apoptosis or autophagy (Fig. 4).

As illustrated by dendrogram and PCA (Figs. 5 and 6), difference in the secretomes of differentiated SH-SY5Y and normal SH-SY5Y cells confirmed that stimulation with BDNF after the initial RA-treatment led to a change in protein machinery and secretion of SH-SY5Y cells. These results further consolidate the conclusion that the normal SH-SY5Y cells can be an *in vitro* model of addressing specific questions related to the pathobiology of AD.

Differentiated SH-SY5Y cell secretome was clearly discriminated from KLVFF, RGKLVFFGR and RIIGL differentiated treated cells (Figs. 5 and 6). A reason for such clustering can be caused by different response of differentiated SH-SY5Y cells after exposure to the peptides. The KLVFF secretome was closely distributed around the origin of the score plot, while the RGKLVFFGR secretome was influenced in the negative direction of PC2. The different trends observed for KLVFF and (RG)KLVFF(GR) secretomes can be explained by the presence of arginine (R) and glycine (G) at both ends of KLVFF. This alteration in net charge, structure, and amino acid composition of (RG)KLVFF(GR) peptide resulted in a higher resistance to proteolysis, higher cytotoxicity against differentiated SH-SY5Y cells, and lower effect on the viability of normal SH-SY5Y cells. This finding is in agreement with *Austen et al. (2008)*.

Of the 290 proteins detected in secretome of RIIGL treated differentiated SH-SY5Y cells, a total of 5 proteins were uniquely detected (Fig. 7). They are glial cell line-derived neurotrophic factor (GDNF), transmembrane emp24 domain-containing protein 10 (TMED10), putative inactive neutral ceramidase B (ASAH2B), NADH-ubiquinone oxidoreductase chain 4 (MT-ND4) and neurogranin (NRGN). The protein interactions network of ASAH2B, GDNF, MT-ND4, TMED10 showed the strong relationship with AD related proteins including MAPT and APP using STITCH 5.0 as shown in Fig. 8.

Glial cell-line derived neurotrophic factor (GDNF) was proposed to play a role in neurodegenerative disease especially damage of cholinergic CNS neurons like AD. Lower GDNF levels in the blood and higher levels in cerebrospinal fluid samples from AD patients suggest that GDNF may be a potential target protein for AD-related pathology (*Straten et al., 2009*). Moreover, the stimulation of the PI3K/AKT pathway by GDNF promotes neuronal survival by making cells resistant to apoptosis (*Villegas et al., 2006*).

Transmembrane emp24 domain-containing protein 10 (TMED10) expression was found to localize at the presynaptic membranes of neuronal junctions and plays a role in Aβ secretion (*Laßek et al., 2013*; *Liu, Fujino & Nishimura, 2015*). In addition, TMED10 can suppress the transport of amyloid precursor proteins against the secretory pathway, resulting in less accumulation of amyloid precursor protein. Therefore, TMED10 might involve in prevention of AD pathogenesis in humans (*Liu, Fujino & Nishimura, 2015*; *Chen et al., 2006*).

Ceramidase deacylates ceramide was found to lead to the production of sphingosine and a free fatty acid (*Rohrbough et al., 2004*). Neutral ceramidase B localizes largely to the plasma membrane (*Mao & Obeid, 2008*). An increased membrane-associated oxidative stress and excessive production and accumulation of ceramides in AD were reported (*Haughey et al., 2010*). Perturbed sphingomyelin metabolism by altering level of inactive neutral ceramidase B (ASAH2B) may be related to the dysfunction and degeneration of neurons that occur in AD.

Expression of NADH-ubiquinone oxidoreductase chain 4 (MT-ND4) is essential for NADH: ubiquinone oxidoreductase complex I activity (*Hofhaus & Attardi, 1993*). MT-ND4 defects are associated with aging and neurodegeneration including AD (*Chou et al., 2011*; *Rhein et al., 2009*; *Zhang et al., 2015*). RIIGL peptide may bind to the redox center of complex I leading to elevated AMP/ATP ratio and activate AMP-activated protein kinase

in differentiated SH-SY5Y cells without inducing oxidative damage or inflammation. It suggests that metabolic reprogramming induced by modulation of mitochondrial complex I activity represents promising therapeutic strategy for AD.

Neurogranin (NRGN) is a postsynaptic protein playing role in long-term potentiation and synaptic plasticity (*Gerendasy & Sutcliffe, 1997*; *Kaleka & Gerges, 2016*). Neurogranin is a redox-sensitive protein and is a likely target of nitric oxide (NO) and other oxidants (*Sheu et al., 1996*). The secretory neurogranin detected in differentiated SH-SY5Y cells might response to oxidative stress induced by RIIGL peptide.

The results indicate that an increase in these five proteins is an important mechanism in the action of RIIGL peptides on cognitive function. The RIIGL peptide acts on the SH-SY5Y cells by regulating amyloid-beta formation, neuron apoptotic process, ceramide catabolic process and oxidative phosphorylation. The study shows that the peptide has a high potential to treat AD.

A primary limitation of our study was the absence of protein expression validation, which could have enhanced the study's reliability. In recent years, the use of methods such as ELISA or western blot for validating proteomic data has been criticized. Concerns related to antibody specificity and availability have been raised, and the development of highly specific antibodies remains a significant challenge. Additionally, differences in sample preparation between proteomics and other techniques may contribute to varying results (*Handler et al., 2018*; *Mann, 2008*; *Mehta et al., 2022*; *Nakayasu et al., 2021*). Furthermore, the lack of available antibodies targeting specific proteins within the context of our study posed a substantial issue. Unfortunately, the high cost and time-intensive nature of creating new antibodies hindered our ability to conduct a comprehensive validation study. We will incorporate these points into the discussion section of the manuscript.

## CONCLUSIONS

RIIGL peptide has been shown to have a neuroprotective effect on SH-SY5Y cytotoxicity induced by retinoic acid (RA) and brain-derived neurotrophic factor (BDNF). In an AD *in vitro* model, five proteins, namely ASAH2B, GDNF, MT-ND4, and TMED10, were uniquely detected after exposure to RIIGL peptide. The interaction network between these five proteins and AD-related proteins was also observed. Further studies are recommended to modify the peptide or combine it with other active agents to reduce toxicity and enhance its activity.

## ACKNOWLEDGEMENTS

We would like to thank staff member of Functional Proteomics Technology Laboratory, National Center for Genetic Engineering and Biotechnology, National Science and Technology Development Agency for kind supports in providing technical material, equipment and cell culture.

### Funding

This work was funded by the Thailand Science Research and Innovation (TSRI) Fundamental Fund fiscal year 2022, Grant number: 159271, Royal Thai Government Grant supported by Thammasat University. There was no additional external funding received for this study The funders had no role in study design, data collection and analysis, decision to publish, or preparation of the manuscript.

### Grant Disclosures

The following grant information was disclosed by the authors:
Thailand Science Research and Innovation: 159271.
Thammasat University.

### Competing Interests

The authors declare there are no competing interests.

### Author Contributions

- Sittiruk Roytrakul conceived and designed the experiments, performed the experiments, analyzed the data, prepared figures and/or tables, authored or reviewed drafts of the article, and approved the final draft.
- Janthima Jaresitthikunchai performed the experiments, analyzed the data, prepared figures and/or tables, and approved the final draft.
- Narumon Phaonakrop performed the experiments, analyzed the data, prepared figures and/or tables, and approved the final draft.
- Sawanya Charoenlappanit performed the experiments, analyzed the data, prepared figures and/or tables, and approved the final draft.
- Siriwan Thaisakun performed the experiments, analyzed the data, prepared figures and/or tables, and approved the final draft.
- Nitithorn Kumsri performed the experiments, prepared figures and/or tables, and approved the final draft.
- Teerakul Arpornsuwan conceived and designed the experiments, performed the experiments, analyzed the data, authored or reviewed drafts of the article, and approved the final draft.

### Data Availability

The MS/MS raw data and analysis are available in the ProteomeXchange Consortium via the jPOST partner repository: JPST002316 and PXD045244.

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
