# Peer review of "Secretomic changes of amyloid beta peptides on Alzheimer’s disease related proteins in differentiated human SH-SY5Y neuroblastoma cells"

_PeerJ, doi:10.7717/peerj.17732_

## Round 0.1 · original submission · Major Revisions

This manuscript has several issues in its current form and therefore it needs major revision before acceptance in the journal PeerJ.

Both reviewers' comments need to be addressed.

In addition,

1: for cytotoxicity assay, MTT is an indirect method, authors are suggested to use more specific assays like Lactate dehydrogenase (LDH) assay or CellTox or Propidium iodide staining, etc.

2: Authors should present their cytotoxicity data as a graph clearly showing the number of experiments used to generate the data.

3: Statistical analysis needs to be thoroughly checked before submitting a revised manuscript.

**Language Note:** The review process has identified that the English language must be improved. PeerJ can provide language editing services - please contact us at [email protected] for pricing (be sure to provide your manuscript number and title). Alternatively, you should make your own arrangements to improve the language quality and provide details in your response letter. – PeerJ Staff

Reviewer 1 ·

Basic reporting

No specific comments, please see additional comments.

Experimental design

No specific comments, please see additional comments.

Validity of the findings

No specific comments, please see additional comments.

Additional comments

This study identified the secretomics effects of three synthesized peptides, especially RIIGL, on an in vitro AD model. The results demonstrated that RIIGL peptide could be an effective neuroprotective effect on SH-SY5Y cytotoxicity. Moreover, five proteins, which are closely connected to other AD-related proteins, were uniquely detected after exposure to RIIGL peptide. Overall, the findings are interesting, but there the writing is plain. I would recommend a major revision by polishing the manuscript, as well as addressing the following points before publication.
1. Please check if the tested peptide is KLVFF or KLVF. E.g., line 96, 191, etc.
2. The Vero cell line was initiated from the kidney tissue. What is the significance of selecting this cell line?
3. All the proteins discussed in line 230-232 should be included in Table 3.
4.In Figure 1, why there are triplicate data for each peptide? The corresponding significance should be included in figure caption.
5. The experimental procedures (e.g., western blot or others) are recommended to be included to confirm the proteins extracted cell culture media of SH-SY5Y treated with different peptides.
6. The writing is always confusing. For example, what is the meaning for the sentences from line 337 to line 342? It is hard to catch the author’s point.

Reviewer 2 ·

Basic reporting

In the paper entitled :"Secretomics effects of amyloid beta peptides on Alzheimer’s disease related proteins in differentiated human SH-SY5Y neuroblastoma cell", the authors tried to investigate the effects of 3 synthetic peptides on an AD in vitro model represented by differentiated SH-SY5Y neuroblastoma cells exposed to retinoic acid (RA) and brain-derived neurotrophic factor (BDNF). Although the research is well designed, there many are various weakness on the paper.

1- The authors should go through the entire paper and make the english better.

2- Thee abstract is long and not well structured.

3- The introduction should be more organized and better referenced. for example the authors can discuss the different therapeutic strategies briefly before they talk about the three peptides that they selected. Also the authors didn't talk about the different peptides that showed effect on AD which in consequence make unclear to the readers: why the authors selected these three specific peptides among others???

4- The reference list should be updated since many relevant references were not included

3- Line 66 and 65: telling that there is no currently available effective treatment and giving a reference dared 2020, is not completely true. There are many advances since 2020 mainly on the monoclonal antibodies.

Experimental design

1- The paper lack originality. Why the authors used these three peptides that were already known to have an effect on AD. Some of these peptides was used since 2004? Where is the Novelty of using old peptides already known for their effects?

Methods Line 96: More details should be added to the peptide synthesis and purification. Also using 85% purity is not the best to use in order to have 100% confirmative results.

Validity of the findings

Again the Novelty of using three old peptides to study their effect on AD is questionable? The authors should explain their choice of the use of these peptides among others

Additional comments

The authors should work more on their discussion that should be more organized and with deeper information and more critical thinking

---

## Round 0.2 · accepted · Accept

The authors have comprehensively addressed all of the reviewers' comments and implemented the necessary revisions to improve the manuscript's clarity and scientific merit. This manuscript is ready for the publication.

Reviewer 1 ·

Basic reporting

All included in additional comments

Experimental design

All included in additional comments

Validity of the findings

All included in additional comments

Additional comments

Thanks for addressing all my concerns.

Reviewer 2 ·

Basic reporting

The authors addressed all my previous comments and suggestions.

Experimental design

The authors addressed all my previous comments and suggestions.

Validity of the findings

The authors addressed all my previous comments and suggestions.

Additional comments

The authors addressed all my previous comments and suggestions.